# Evaluation of Pharmacokinetic and Toxicological Parameters of Arnica Tincture after Dermal Application In Vivo

**DOI:** 10.3390/pharmaceutics14112379

**Published:** 2022-11-04

**Authors:** Franziska M. Jürgens, Sara M. Robledo, Thomas J. Schmidt

**Affiliations:** 1Institute of Pharmaceutical Biology and Phytochemistry, University of Münster, Corrensstrasse 48, D-48149 Münster, Germany; 2PECET-School of Medicine, University of Antioquia, Calle 70 # 52-21, Medellin 0500100, Colombia

**Keywords:** Arnica tincture, sesquiterpene lactones, helenalin, dermal absorption, dermal toxicity

## Abstract

Cutaneous leishmaniasis (CL) is classified as a neglected tropical disease by the World Health Organization. As the standard drugs for the treatment of this disease suffer from severe unwanted effects, new effective and safe therapeutic options are required. In our previous work, Arnica tincture showed promising antileishmanial effects in vitro and in vivo. For the potential treatment of human CL patients with Arnica tincture, data on the pharmacokinetic properties of the bioactive, antileishmanial compounds (the sesquiterpene lactone (STL) helenalin and its derivatives) are needed. Therefore, we studied the in vivo absorption of the bioactive compounds after the dermal application of Arnica tincture in rats. Moreover, we analyzed the blood plasma, urine, and feces of the animals by ultra-high-performance liquid chromatography coupled to high-resolution mass spectrometry (UHPLC-HRMS). Although the majority (84%) of the applied STLs (1.0 mg) were absorbed, the concentrations in the plasma, urine, and feces were below the limit of detection (0.3 ng/mL) in the samples for UHPLC-HRMS analysis. This result may be explained by extensive metabolism and slow permeation accompanied by the accumulation of STLs in the skin, as described in our previous work. Accordingly, the plasma concentration of STLs after the topical application of Arnica tincture was very far from a dose where toxicity could be expected. Additionally, tests for corrosive or irritant activity as well as acute and repeated-dose dermal toxicity did not show any positive results after the administration of the amounts of Arnica tincture that would be needed for the treatment of CL. Consequently, in the treatment of CL patients with Arnica tincture, no toxic effects are expected, other than the known sensitization potential of the STLs.

## 1. Introduction

The neglected tropical disease cutaneous leishmaniasis (CL) is caused by parasites of the genus *Leishmania*, which cause serious skin lesions that tend to ulcerate, are slow to heal, and may lead to severe disfiguration, leaving life-long scars [1]. Although more than one million new cases occur annually, only a few therapeutic options are available [2]. The standard treatment with antimony-based drugs suffers from toxic effects, high costs, and low efficacy for some *Leishmania* species [3]. Moreover, the treatment is carried out with intralesional injections that must be performed by trained medical staff and is rather unpleasant for the patient [4]. Consequently, new, efficacious, safe, affordable, and easy-to-administer drugs are urgently needed. We recently reported on the promising efficacy of Arnica tincture in the treatment of CL in golden hamsters [5]. Arnica tincture is an ethanolic tincture prepared from the flowerheads of *Arnica montana* L., a medicinal plant traditionally used for the treatment of bruises, sprains, and localized muscle pain [6]. The tincture is an approved herbal medicinal product in Europe, and its preparation is prescribed in the European Pharmacopoeia [7]. The main active substances are sesquiterpene lactones (STLs, i.e., esters of helenalin (H) and 11α,13-dihydrohelenalin (DH)), which have anti-inflammatory as well as antileishmanial activities [8,9]. Before attempting to use Arnica tincture as a CL treatment in a study with human patients, we investigated the pharmacokinetic properties of its STLs after dermal application.

In previous communications, we reported on the in vitro metabolism and the ex vivo dermal absorption of the STLs from Arnica tincture [10,11]. A fast metabolism, dominated by glutathione conjugation, and an extensive absorption of the STLs into the skin were observed with only a low degree of permeation. Thus, an accumulation in the skin was found. The distribution and elimination of a radiolabeled Arnica STL ([^3^H]-DH) after intraperitoneal (i.p.) and intravenous (i.v.) applications had previously been studied by Grippo et al., who found some accumulation in the muscle, bone, and adipose tissue as well as the skin [12]. In the present study, we therefore investigated the dermal absorption of STLs from Arnica tincture in vivo.

Another important aspect to take into account before applying a drug in human patients is its toxicity. To date, only a few reliable data on the dermal toxicity of Arnica are available in the literature. For a dermally applied Arnica extract, it was not possible in tests with rabbits to determine the dose at which 50% of the test animals die (LD_50_). The highest tested dose was 5 g/kg bodyweight (bw) [13]. Data on the toxicity of individual STLs are only available for nondermal routes of administration. The LD_50_ values of helenalin were 43 mg/kg bw (mouse) after i.p. administration [14] and 85–150 mg/kg bw after p.o. administration (mouse, rat, hamster, rabbit, and sheep) [15].

The dermal application of Arnica tincture is considered safe by the European Medicines Agency (EMA) [16]. Various clinical trials with Arnica tincture for the topical treatment of disorders related to the traditional use (see above) were carried out so far [16]. Some of these studies reported on the formation of erythema and on a potential for sensitization accompanied by the development of contact dermatitis [13,16]. Different Arnica extracts were tested for phototoxicity, but no positive results were found in any of the experiments [16]. The studies on irritant activity performed to date (overview in [13]) have not been conducted in accordance with the Organization of Economic Cooperation and Development (OECD) guidelines. Therefore, we found it necessary to perform a new study to test the irritant effects, which was designed according to the OECD guideline (no. 439) [17]. In addition, a possible corrosive effect, the acute dermal toxicity, and the repeated-dose dermal toxicity were studied in accordance with the respective OECD guidelines [18,19,20].

## 2. Materials and Methods

### 2.1. Chemicals and Materials

Chemicals were purchased from Merck KGaA (Darmstadt, Germany), Thermo Fisher Scientific (Schwerte, Germany), or Carl Roth GmbH and Co. KG (Karlsruhe, Germany). The Arnica tincture from the manufacturer Hetterich (batch numbers 144603 and 144901) was purchased from a local pharmacy. Oasis HLB SPE cartridges were purchased from Waters Corporation (Milford, MA, USA). Distilled water, phosphate-buffered saline (PBS), and sodium dodecyl sulfate (SDS) were purchase from Invitrogen-Thermo Fisher Scientific (Waltham, MA, USA). Potassium hydroxide (KOH) was purchased from HACH (Hach, Colombia). The 3-(4,5-dimethylthiazol-2-yl)-2,5-diphenyltetrazolium bromide (MTT) was purchased from Sigma-Aldrich (St. Louis, MO, USA). SkinEthic^TM^ (Episkin, France) was used as a reconstructed human epidermis (RhE).

### 2.2. In Vivo Dermal Absorption and In Vivo Pharmacokinetics

One dermal absorption study and two pharmacokinetic studies (PK1 and PK2) were conducted with male Wistar rats with an average weight of 555 g. The Arnica groups consisted of 20 animals (dermal absorption study), 8 animals (PK1), or 5 animals (PK2). The control groups included 10 animals (dermal absorption study), 4 animals (PK1), and 2 animals (PK2). The application area (10 cm² in the dermal absorption study and 21.6 cm² in the PK studies) was carefully shaved 24 h before the application. Arnica tincture (0.2 mL in the dermal absorption study and 2.0 mL in PK study) was slowly applied to the shaved skin area. The control groups received the corresponding volume (0.2 mL or 2.0 mL) of PBS. The treated area was covered with a nonocclusive gauze during the exposure period so that the test substances were not removed or ingested orally. After 6 h, the treated skin was washed three times with 10.0 mL of distilled water (skin wash solution). Blood plasma samples were taken before the application (0 h) and after 6 h, 24 h, 48 h, 72 h, and 96 h. Urine and feces were collected after 24 h, 48 h, 72 h, and 96 h. At the end of the dermal absorption study, samples of the stratum corneum were taken with sticky tapes, whereas in the PK studies the complete skin areas of the previously euthanized animals were removed for the STL analysis.

### 2.3. Irritant and Corrosive Activity Tests

The irritant and corrosive effects of Arnica tincture were investigated in triplicate according to OECD guidelines 431 and 439 [17,19], respectively, using reconstructed human epidermis (RhE [17]). PBS was used as a negative control, and 20% SDS (irritant effect) or 8N KOH (corrosive effect) were used as positive controls. To test the irritant effect, the incubation with Arnica tincture (20 µL, corresponding to 9.6 µg of STLs, applied to 0.38 cm^2^ of RhE) lasted 42 min, after which the skin samples remained in the medium without the test substance for 42 h. To test the corrosive effect, the skin samples were incubated with Arnica tincture (50 μL, corresponding to 24 µg STL, applied to 0.38 cm^2^ of RhE) for 3 min and 60 min. Subsequently, the viability was determined by MTT assay as described in [5]. In addition, histological examinations of the treated samples were performed.

### 2.4. Acute Dermal Toxicity and Repeated-Dose Dermal Toxicity Studies

Acute as well as repeated-dose dermal toxicity were studied with Swiss albino mice. One day prior to the treatment, all dorsal and flank fur was removed under inhalation anesthesia. Repeated-dose exposure lasted 21 days, and acute toxicity exposure lasted 24 h. Mice in the control group (*n* = 4 in the acute toxicity group or *n* = 6 in the repeated-exposure group) were only shaved weekly, without any substance being applied. Mice in the Arnica groups (*n* = 6 for acute toxicity or *n* = 10 for repeated exposure) received 200 μL (single dose) or 80 μL (repeated dose) of Arnica tincture evenly applied to the depilated skin area. The treated area was covered throughout the exposure period with gauze and nonirritating adhesive tape to ensure that the test substance came into contact with the skin and that the animals could not ingest the substance orally. After the treatment, the gauze was removed, and the residues of the test substance were washed away with sterile water.

The animals were monitored each day for behavioral changes or changes in the treated skin area, the fur, eyes, and mucous membranes and were weighed every week. To take blood samples, the mice were anesthetized with ketamine and xylazine. At the end of the study, all mice were euthanized in a CO_2_ chamber, necropsy was performed, and a skin fragment was collected for histopathological analysis.

### 2.5. STL Extraction

STLs were extracted from the skin wash solutions, plasma, and urine using Oasis hydrophilic–lipophilic balance solid-phase extraction (SPE) cartridges with 30 mg of sorbent quantity and a 1 mL capacity (Waters GmbH, Eschborn, Germany). Plasma and urine samples were previously heated to 65 °C for 60 min and centrifuged for 1 min at 10,000 rpm to remove proteins. The supernatant was then diluted 1:1 with water to reduce the viscosity of the samples. The cartridges were placed on an SPE vacuum chamber, conditioned with 0.5 mL of methanol, and equilibrated with 0.5 mL of water. Next, samples (approx. 50 µL of plasma, 0.5 mL of urine, or 2.0 mL of PBS) mixed with 10 µL of the internal standard, α-santonin (10 µg/mL), were loaded on the cartridges. After washing twice with 0.2 mL of water, the analytes were eluted twice with 0.2 mL of acetonitrile/methanol (90/10, *v*/*v*). The solvent was evaporated by vacuum centrifugation, and the samples were dissolved in 1.0 mL of water/acetonitrile (95/5, *v*/*v*) to suit the UHPLC starting conditions.

Feces samples were freeze-dried for 24 h and then pestled. The dry, pulverized feces samples weighed 0.1–0.3 g. To each sample, 7.0 mL of methanol with 0.1 µg of α-santonin, as internal standard, were added. Subsequently, samples were vortex-mixed (1 min), shaken for 1 h at 320 rpm, and sonicated for 5 min. After centrifugation (5 min at 4000 rpm), 5.0 mL of the supernatant was replaced with 5.00 mL of methanol. This extraction was performed four times. Next, the combined methanol phases were evaporated by vacuum centrifugation, and the residues were dissolved in 1.0 mL of water/acetonitrile (95/5, *v*/*v*), centrifuged, and transferred to HPLC vials.

To the gauze (72–99 cm²), 10.0 mL of methanol and 0.1 µg of α-santonin (internal standard) were added. Extraction took place by shaking (1 h at 200 rpm) and sonification (15 min). The supernatant was replaced with 5.0 mL of methanol, and the process was repeated. After four extractions, the methanol was combined and reduced to 90 µL by vacuum centrifugation.

Skin samples (approx. 20 cm²) were cut in pieces of 1.0–1.5 cm² and placed in 15 mL reaction tubes. To the samples, 5.0 mL of methanol with 0.1 µg of the internal standard (α-santonin) were added. Subsequently, samples were shaken on a laboratory shaker (1 h at 400 rpm), sonicated (5 min), and centrifuged (5 min at 4000 rpm). Then, 4.0 mL of the supernatant was replaced with methanol. After four extractions, the supernatants were combined, reduced to 1.0 mL, and purified with a Sephadex LH-20 column (2.0 cm × 1.0 cm). The sorbent was preswelled for at least 24 h in methanol, slurried, and filled into the column. Samples were loaded on the column and eluted with 3.0 mL of methanol. The eluate was evaporated, and the residues were dissolved in 1.0 mL of water/acetonitrile (95/5, *v*/*v*). Likewise, the adhesive tapes containing stratum corneum were transferred to reaction tubes, and 10.0 mL of methanol and 0.1 µg of the internal standard (α-santonin) were added. After shaking on a laboratory shaker (4 h at 320 rpm), the extraction agent was removed, reduced to 1.0 mL by vacuum centrifugation, and purified using a Sephadex LH-20 column (2.5 cm × 0.5 cm), as described above. The samples were loaded on the column and eluted twice with 1.0 mL of methanol each. The eluate was reduced to 90 μL of methanol by vacuum centrifugation and transferred to an HPLC vial with an insert.

Arnica tinctures were prepared for analysis by adding 100 µL of the internal standard, α-santonin (10 µg/mL), and 761 µL of methanol to 139 µL of each tincture.

### 2.6. STL Quantification

The UHPLC-HRMS analysis of urine, plasma, skin wash solutions, feces, gauze, skin samples, and Arnica tinctures was carried out with the method described in [11]. Briefly, the separation was performed with an RP-18 column (Dionex Acclaim RSLC 120 C18, 0.1 m × 2.1 mm × 2.2 μm, Thermo Fisher Scientific, Inc., Waltham, MA, USA) at 40 °C. A gradient of water and acetonitrile with 0.1% formic acid was used. After ionization in positive electrospray mode (4.5 kV), a mass range of *m*/*z* 50–1500 was transferred to the time-of-flight tube. The calibration was accomplished with a solution of sodium formate (10 mM) in propan-2-ol/water/formic acid/sodium hydroxide (250/250/1/5, *v*/*v*/*v*/*v*). For the operation of the system, data acquisition, data processing, and evaluation, the software package Compass 1.5 and the programs Chromeleon and Metabolite Detect were used. Difference chromatograms were generated with the eXpose algorithm (ratio = 5.00, retention time difference ± 0.20 min, m/z difference ± 0.10).

## 3. Results

### 3.1. In Vivo Dermal Absorption Study

To verify that our previous ex vivo dermal absorption experiments [11] reflected the in vivo conditions, an in vivo dermal absorption study was performed with healthy Wistar rats. Arnica tincture (200 µL, corresponding to 96.1 µg of STLs) was applied to 10 cm² of skin and covered with gauze for 6 h. Subsequently, the skin was washed with phosphate-buffered saline (PBS, skin wash solution), and the liquid was collected to determine the amount of STLs remaining on the skin’s surface. In addition, STLs were extracted from the gauze and quantified (Table 1).

An average of 12.1 µg of STLs (corresponding to 13% of the total dose) was quantified in the skin wash solutions. The gauze contained an average of 3.22 µg of STLs (corresponding to 3.3% of the total dose). The amounts in the skin wash solutions and in the gauze together represented the nonabsorbed fraction (15.4 µg or 16%). From this, it could be concluded that approximately 80.7 µg (84%) of the applied dose (96.1 µg) was absorbed. Next, the proportion of the absorbed fraction in the stratum corneum was determined. Sampling was carried out with adhesive strips 6 h (animals 1–4), 24 h (animals 5–8), 48 h (animals 9–12), 72 h (animals 13–16), or 96 h (animals 17–20) after the application of the tincture. Interestingly, STLs could only be detected in the stratum corneum after 6 h, i.e., directly after the removal of the Arnica tincture by skin washing. In these samples, a small STL amount of 0.29 ± 0.07 µg (0.3 ± 0.1% of the total amount) was quantified. To investigate STL uptake into the blood (resorption) as well as elimination, samples of blood, urine, and feces were collected after 6 h, 24 h, 48 h, 72 h, and 96 h. However, no STLs could be detected in these samples.

### 3.2. In Vivo Pharmacokinetic Study

In order to analyze the STLs in blood plasma, urine, and feces, two independent pharmacokinetic studies were performed with an increased volume of Arnica tincture (2.0 mL) compared to the volume used in the in vivo dermal absorption study (0.2 mL). In one study (PK1), eight animals were treated with Arnica tincture (Arnica group) and four animals were treated with PBS (control group). In the other study (PK2), five animals were in the Arnica group and two animals were in the control group. STL concentrations of 481 µg/mL and 524 µg/mL were quantified for the Arnica tincture (two different batches) used in PK1 and PK2, respectively. The test substance was applied topically to shaved skin areas of 21.6 cm², covered with gauze for 6 h, and then washed off. The skin wash solution was collected, and the nonabsorbed STL amount was determined. Blood samples were taken from each animal before treatment (0 h) and after 6 h, 24 h, 48 h, 72 h, and 96 h. Urine and fecal samples were collected after 24 h, 48 h, 72 h, and 96 h. Since, in the in vivo dermal absorption study, STLs were only quantifiable in the stratum corneum samples taken after 6 h, the entire treated skin area was removed and extracted at the end of the PK studies (after 96 h). This method allowed STL quantification in the skin of all animals (Table 2). In PK1, the STL content in the skin samples averaged 0.006% of the total applied amount, while in PK2 it was 0.037%. In the skin wash solutions, 7.7% (PK1) and 4.2% (PK2) of the total amount was quantified.

In addition, 114 plasma, 76 urine, and 76 fecal samples were extracted and analyzed by UHPLC-HRMS, as described in Section 2.5 and Section 2.6. Extracted ion chromatograms (EICs) of the analyte backbones were generated, in particular *m*/*z* 245.1172 (Hs) and *m*/*z* 247.1329 (DHs) (see [11]). Further, EICs for the expected metabolites were created based on the observed in vitro hepatic metabolites and ex vivo dermal metabolites in our previous studies [10,11] (e.g., hydroxides, water adducts, glutathione conjugates, cysteinylglycin (CysGly) conjugates, cysteine (Cys) conjugates, and mercapturic acids (NAC)). Additionally, difference chromatograms of the Arnica and control samples were generated and evaluated. However, despite an exhaustive search, neither the STLs nor their metabolites could be detected using these approaches.

### 3.3. Toxicity Studies

Since Arnica tincture is applied dermally, potential corrosive and irritant effects, acute dermal toxicity, and repeated-dose dermal toxicity were tested.

#### 3.3.1. Corrosive Activity

Reconstructed human epidermis (RhE), as specified in the OECD guidelines no. 431 and no. 439 on corrosive and irritant activity [17,19], was used to test the corrosive effect of Arnica tincture. Briefly, RhE is a highly differentiated multilayer epidermis model that is derived from the cultivation of human-derived epidermal keratinocytes. It consists of basal, spinous, and granular layers, a multilayered stratum corneum, and intercellular lipids [17]. The results are shown in Figure 1. The viability after incubation with the negative control (PBS, C-) was normalized to 100%. Incubation with the corrosive positive control (8N KOH, C+) resulted in 37.6% viability after 3 min and 21.5% viability after 60 min. When incubated with Arnica tincture (50 μL, corresponding to 24 µg of STLs, applied to 0.38 cm^2^ of RhE), viability was 92.6% after 3 min and 83.4% after 60 min.

In histological examinations of the treated samples, tissue atrophy was diagnosed in the samples treated with the positive control since no stratum granulosum was present. In contrast, cell layers of normal epithelium including stratum granulosum were detected in the samples of the negative control and those treated with Arnica tincture. Therefore, Arnica tincture was classified as noncorrosive, according to the European Union classification R38.

#### 3.3.2. Irritant Activity

In order to test for potential irritant activity, a sample of RhE (0.38 cm^2^) was exposed to the test compound for 42 min. Subsequently, the skin was washed, and the medium was replaced by medium without the test compound. The skin was then incubated for 42 h in order to determine any long-term tissue damage after exposure to the test compound. The results are shown in Figure 2. The viability after incubation with the negative control (PBS, C-) was normalized to 100%. Skin tissue samples exposed to Arnica tincture (20 μL, corresponding to 9.6 µg of STLs) yielded a viability of 93.9% after 42 min of exposure and 42 h of incubation, while exposure to the positive control (20% sodium dodecyl sulfate, C+) under the same conditions yielded a viability of only 18.5%.

In histological analyses of tissues exposed to PBS (negative control) and Arnica tincture, cell layers corresponding to the granular and corneal layers typical of a normal epithelium were found. For the tissue exposed to the positive control, an atrophic spinal layer was found with no granular layer and subepithelial dermis. The diagnosis in this positive control tissue was atrophy. In conclusion, Arnica tincture can be considered nonirritant, according to European Union classification R38.

#### 3.3.3. Acute Dermal Toxicity

In acute dermal toxicity testing, all fur was removed from the dorsal/flank area one day before the application of the test substance. Arnica tincture (200 µL, equivalent to 96.1 μg of STLs) was uniformly applied to the depilated skin areas (dorsal/flank) of six mice and covered with porous gauze and nonirritating tape for a 24 h exposure period. Four other mice belonged to a control group, which was only shaved, without any test substance being applied. After 24 h, the gauze was removed, and residual Arnica tincture was washed off with sterile water. All animals were closely observed from the treatment day (TD 0) until post-treatment day (PTD) 14 (the end of the study) and were weighed every week. No alterations in the behavior of the animals were detected during the study. The weight of all animals was stable and did not show any significant difference from healthy animals (Table 3).

No signs of irritation or other signs of skin toxicity associated with exposure to the Arnica tincture were observed at any time during the study. Some erythematous lesions were observed in one of the treated animals at PTD 8 and PTD 14 and two mice of the control group at PTD 1. Nonetheless, these lesions were associated with irritation due to self-injury due to scratching. In addition, the lesions appeared outside the application area (Appendix A, animals marked with *). No deaths were recorded in any of the groups. All together, these results confirm that a single exposure of Arnica tincture is safe for animal skin, as observed for 14 days after exposure.

#### 3.3.4. Repeated-Dose Dermal Toxicity

One day before the start of the study, all fur was removed from the dorsal/flank area. Arnica tincture (80 μL, equivalent to 38.4 μg of STLs) was applied daily to the skin area for 21 days. Control mice were shaved weekly without the application of any compound. Animals were followed-up for two weeks after the end of treatment. Every day, animals were observed for any behavioral changes or alterations in the treated skin, fur, eyes, and mucous membranes. The body weight was recorded every week from TD 0 to PTD 14. A picture of every animal’s skin was taken (Appendix A). Weight gains were normal and stable in all mice at all evaluation times (Table 4).

Moreover, blood samples were taken and analyzed (Table 5). Treated animals and controls had normal values of alanine aminotransferase (ALT), alkaline phosphatase (AP), creatinine, and blood urea nitrogen (BUN). Further, hemoglobin (Hb), hematocrit (Hto), red blood cell counts (RBC) and white blood cell counts (WBC) were all within the normal ranges.

During necropsy, no macroscopic alterations were found in any organ or tissue. Microscopic alterations in the skin were also not observed. A histological analysis of the skin samples showed no lesions of any kind that could suggest any type of toxicity. Taken together, these results confirm that Arnica tincture is safe for animal skin after 21 days of daily exposure.

## 4. Discussion

In this article, the in vivo absorption of Arnica STLs and their toxicity after the dermal application of Arnica tincture are reported. First, the results of the in vivo dermal absorption study are compared with the results of our previous ex vivo dermal absorption studies. Skin extracts of the former study contained 0.3% of the applied STL dosage, which was lower than in the ex vivo studies, where 6.0–7.8% of STLs were recovered from skin extracts. However, it should be noted that in the latter study a larger STL fraction of 97.6–99.3% was absorbed within 48 h, rather than 84.0% within 6 h, which could explain the difference. In stratum corneum samples, STLs were detected only when sampling after 6 h, i.e., immediately after skin washing. It is suggested that in the other animals the STLs permeated into deeper skin layers or were bound to skin constituents such as keratin [11] during the time between skin washing (6 h) and stratum corneum sampling (24–96 h). The binding of a large fraction of the STLs to skin constituents is very likely, as we could previously show an accumulation of a fluorescent helenalin ester in the stratum corneum by fluorescence microscopy [11]. In the PK studies, the complete treated skin area was collected so that STLs could still be quantified after 96 h (<0.05% of the total applied dosage). The fact that a larger STL amount was quantified in the skin samples after 6 h (0.3%) compared to 0.05% after 96 h is plausible if one assumes that the unbound fraction of STLs continuously permeated from the stratum corneum into deeper skin layers, was absorbed into the blood, and was thus no longer present in the treated skin site after a longer time period.

The PK studies were performed with male Wistar rats with an average weight of 555 g for which a blood volume of approximately 40 mL can be assumed [22]. No STLs were detected in the plasma, urine, or fecal samples at any of the investigated time points after the dermal application of 2 mL of Arnica tincture containing about 1000 µg of total STLs. The limit of detection for single STLs in our UHPLC-HRMS method was determined to be 0.3 ng/mL [11]. Consequently, the concentration of any single STL in the UHPLC-HRMS samples was <0.3 ng/mL. During sample preparation, plasma samples (sample volume 50 µL) were purified by solid-phase extraction (SPE) and dissolved in 1.0 mL of water/acetonitrile (95/5, *v*/*v*) to enable five UHPLC-HRMS measurements (injection volume = 200 µL). This corresponded to a dilution of 1:20. With the additional consideration of the recovery of 84%, the STL concentration of the plasma samples must have been <7.0 ng/mL, i.e., <0.5 µg/kg bw. Similarly, for the urine samples with a dilution of 1:2 and a recovery of 82%, an STL concentration of <0.7 ng/mL in the urine could be calculated. For the sample preparation of the fecal samples, the entire sample (max. 0.3 g) was used, purified, and dissolved in 1.0 mL of water/acetonitrile (95/5, *v*/*v*) for UHPLC-HRMS analysis. With a recovery of 80%, the STL concentration was calculated to be <1.3 ng/mL in feces. The maximal concentrations reached in the plasma, urine, or feces by any single STL at any time point after the dermal application of Arnica tincture were thus orders of magnitude below the concentrations reported to be toxic in the case of helenalin (LD_50_ = 43 mg/kg bw, mouse, i.p., with conspicuous toxicity starting at about 8 mg/kg bw [14]). Consequently, the systemic concentrations that were reached can safely be considered nontoxic.

The fate of the STLs after topical application can thus be envisaged as follows (see Figure 3): The volume of Arnica tincture applied per animal (2.0 mL) contained roughly 1000 µg of total STLs (961 µg and 1048 µg in the two separate experiments, PK1 and PK2, respectively), consisting of 22–216 µg and 23–134 µg of the individual STLs of the H and DH types [11]. In the in vivo dermal absorption experiment, it was determined that after 6 h of application, 84.0% of STLs were absorbed into the rat skin. This corresponded to 840 µg divided into 18–181 µg of the individual substances.

In our previous ex vivo absorption experiments with porcine skin, it was found that the major amount of STLs reached the receptor fluid between 24 and 48 h (6.8% of the total applied amount) [11]. A similar absorption behavior is assumed for rat skin. Consequently, the highest STL concentration would have to be in the plasma samples collected after 48 h. Of the STLs absorbed into the skin from 2.0 mL of Arnica tincture, 1–12 µg of the individual substances would have been resorbed into the blood accordingly. In whole blood, the binding of about 75% of the STLs from Arnica tincture was determined by other authors [23], so the theoretical maximum amount of an unbound individual STL in blood would be 3 µg. Furthermore, the results from the metabolism experiments of our previous work [11] showed that the STLs (exemplified by H- and DH-acetates) are extensively metabolized so that after 24 h only 12% of Hs (31–366 ng) and 35% of DHs (88–704 ng) can be expected to be present in an unmetabolized form in the blood. The proportion after 48 h is most likely even lower. In 40 mL of blood, concentrations of 0.8–18 ng/mL would result for the individual STLs (if it were assumed that the compounds would be exclusively present in the blood and not distributed into other compartments). Since only a small blood volume could be collected, each sample consisted of only 50 µL of plasma (containing a maximum absolute amount of 0.04–0.9 ng of each single STL). After SPE purification with a recovery of 84%, the samples were dissolved in a volume of 1.0 mL of water/acetonitrile (95/5). Thus, the theoretical final concentration of each STL in the UHPLC-HRMS samples (approx. 0.04–0.8 ng/mL) was just below the LOD (0.3 ng/mL) for some STLs and just above the LOD for other STLs. Since none of the STLs were detected in plasma, further factors must have had a further decreasing influence on the STL plasma concentration.

As already mentioned, the calculation of the plasma concentration is based on the over-simplified assumption that the unmetabolized STLs would exclusively accumulate in the blood and not be metabolized and/or distributed to other tissues. However, we have shown that rather extensive metabolism takes place during skin penetration [11]. Furthermore, the elimination, i.e., excretion of STLs and/or their metabolites, between blood sampling time points was not taken into account in these considerations. However, it is most likely that a part of the STLs was localized in (fat) tissue, in the urinary bladder, or in the intestine and that STLs were continuously excreted.

It is expected that distribution and excretion of the STLs after resorption into the blood are similar to the results of Grippo et al., who studied these parameters after the i.p. administration of [^3^H]-DH [12]. In this work, accumulations of [^3^H]-DH in the corpus (muscle, bone, and adipose tissue) and in the skin were reported [12], which may thus have contributed to the low plasma concentrations found in the present study. In addition, the distributions of approximately 8% and 58% within 30 min to the liver and intestine observed by the same authors support the assumption that the elimination of the substances was ongoing during our experiments. The fact that there was no detectable STL concentration in either the urine or feces at any of the collection time points is also consistent with continuous excretion. Furthermore, none of the various metabolites expected after our previous investigations could be detected in the excretions. For male Wistar rats, a fecal volume of approx. 21 g/kg bw per day [24] and a urine volume of 60 mL/kg bw can be assumed [25]. This corresponds to 33 mL of urine and 12 g of feces within the 24 h between sample time points, so the excretion volume was comparable to the blood volume (40 mL). Therefore, it seems plausible that the concentrations of either unmetabolized STLs or their various metabolites in urine or feces were well below the limits of detection.

In the toxicity studies with mice and in the PK studies with rats, no signs of toxicity were observed after the dermal application of up to 2 mL of Arnica tincture. For the evaluation of the toxicity of the treatment in humans, the acceptable daily intake (ADI) [26] was calculated. To this end, the “no observed adverse effect level” (NOAEL) [26] is divided by a safety factor, usually 100 [26]. In the repeated-dose dermal toxicity study, mice (20 g bw) were treated daily with 80 µL of Arnica tincture (equivalent to 38.4 µg of STLs) without showing any adverse effects. Thus, an NOAEL ≥ 1.92 mg/kg bw can be derived from this. After dividing this dose by the safety factor of 100, an ADI of 19.2 µg/kg bw is obtained. For an adult human with a 70 kg bw, 1.34 mg of STLs (equivalent to 2.8 mL of Arnica tincture) could safely be applied per day. With this dose, 11.7 cm² of lesion could be treated with 80 µL/cm² three times daily. This is the treatment dose that was proposed on the basis of our in vivo antileishmanial activity study with golden hamsters [5], from which we extrapolated that a treatment with 80 µL/cm^2^ of lesion 3×/day would be suitable for human patients. Assuming a lesion size of 4 cm^2^ and a body weight of 70 kg, the approximate STL amount required for treatment (with a tincture concentration of ≈500 µg STL/mL) would be ≈7 µg STL/kg bw/ day, which is well below the calculated ADI, so this treatment scheme can be considered safe.

Irrespective of these quantitative considerations, it can also be assumed due to the long traditional use of Arnica tincture, that no toxicity is to be expected in humans when the preparation is applied to intact skin. In the case of CL lesions, the uptake of STLs may be higher due to an impaired barrier function. However, the NOAEL mentioned above is only a minimum value since repeated-dose dermal exposure data are known for only one nontoxic dose (80 µL). The NOAEL and ADI may hence actually be much higher. It should be noted that, in the many in vivo experiments with infected golden hamsters [5], not a single observation related to dermal toxicity was made.

With all this being said, it must not remain unmentioned that Arnica preparations have an allergenic potential. A significant number of clinical studies, e.g., for the treatment of chronic venous insufficiency or arthritis pain, have been conducted with Arnica preparations [16]. In rare cases, allergic contact dermatitis (manifested as itching, redness, and the formation of eczema) is known to occur as an adverse effect. Such reactions were reported in approximately 1% of users, especially in persons allergic to other plants of the *Asteraceae* family (cross-sensitivity). The use of herbal preparations of Arnica flowers is considered safe by the EMA, but individuals allergic to Arnica should be excluded from treatments [16]. It should therefore be checked before treatment with Arnica preparations whether the patient has a known allergy to Arnica. Furthermore, the treatment should be stopped if signs of allergy occur. In addition, there are insufficient data for children under twelve years of age, pregnant women, and nursing mothers, so these groups also should not be treated with Arnica preparations as a precautionary measure [16].

## 5. Conclusions

In this article, the results of in vivo dermal absorption, pharmacokinetic, and toxicity studies with STLs from Arnica tincture are reported. The STL amounts extracted from skin in the in vivo studies were comparable to the results of our previous ex vivo dermal absorption study. Moreover, the in vivo dermal absorption experiment confirmed the ex vivo results of extensive STL absorption from dermally applied Arnica tincture into the skin. Nevertheless, the STLs did not reach detectable concentrations in blood plasma, urine, or feces at any of the sampling time points, and we safely conclude that a considerably more sensitive method would be needed to detect such small concentrations. This finding is in agreement with our previous observations that Arnica STLs are absorbed slowly and penetrate the skin only to a low extent, with accumulation in the stratum corneum, combined with extensive metabolism [11]. It may be assumed that the distribution and elimination after dermal absorption are analogous to the i.p. and i.v. applications studied by Grippo et al. [12] once the analytes have been absorbed into the blood, even though some caution may be due since radioactive labeled compounds may behave somewhat different from their unlabeled analogues. For the use of Arnica tincture in the treatment of CL patients by dermal application, the low STL concentrations reached in the plasma, urine, and feces are advantageous, as no toxic effects are to be expected.

Furthermore, no signs of direct acute or chronic dermal toxicity or of irritant or corrosive effects were observed in our studies. Comparing the data on systemic toxicity from the literature (e.g., LD_50_ of helenalin = 43 mg/kg bw, mouse, i.p., see above) and the low doses required for CL treatment (e.g., ≈7 µg/kg bw/day with a lesion size of 4 cm^2^), no toxic effect is to be expected, even when STLs are applied on open wounds. Based on the repeated-dose dermal toxicity study in mice, it was calculated that a lesion area of up to 11.7 cm² in an adult human patient could be treated three times daily with 80 µL of Arnica tincture/cm². In summary, the use of Arnica tincture for the treatment of human CL patients is supported by the results of this work. A first clinical study is under way.

## Figures and Tables

**Figure 1 pharmaceutics-14-02379-f001:**
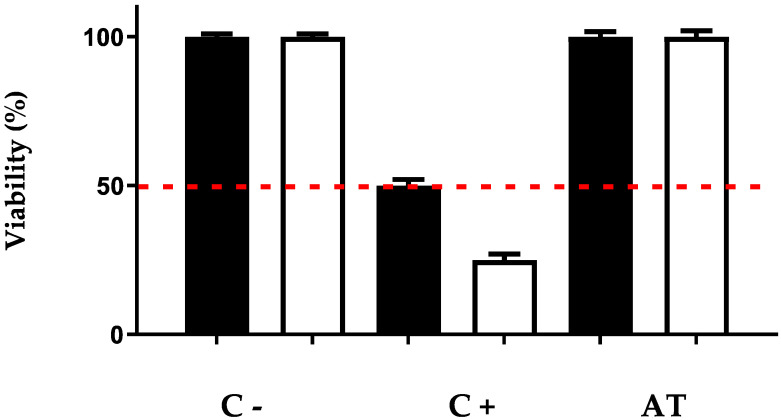
Test for skin corrosive potential of Arnica tincture. The bars represent the means ± SD of the percentage of cell viability after 3 min and 60 min of exposure. C-: negative control; C+: positive control; AT: Arnica tincture. Black bars: 3 min; white bars: 60 min.

**Figure 2 pharmaceutics-14-02379-f002:**
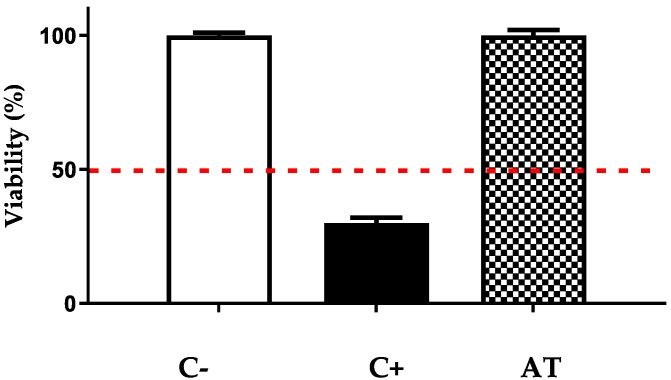
Test for skin irritant potential of Arnica tincture. The bars represent the means ± SD of the percentage of cell viability after 42 min of exposure and 42 h of incubation. C-: negative control; C+: positive control; AT: Arnica tincture.

**Figure 3 pharmaceutics-14-02379-f003:**
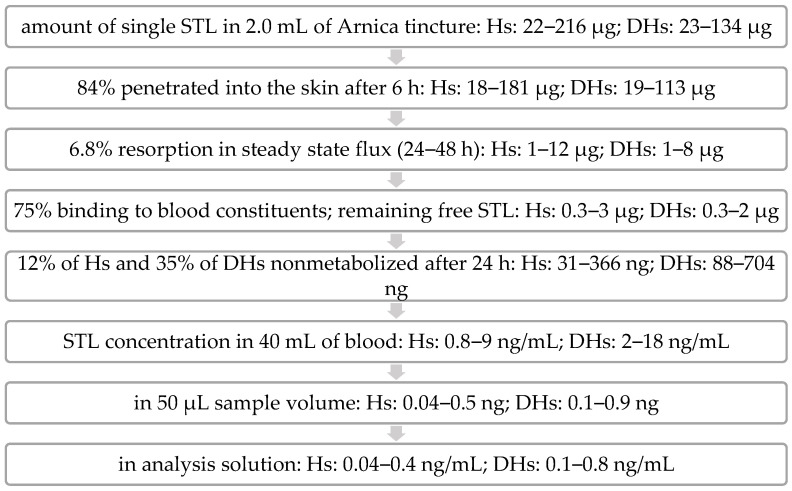
Scheme of theoretical STL concentration in plasma of rats in PK studies.

**Table 1 pharmaceutics-14-02379-t001:** STL amount and percentage of the applied dose (96.1 µg) in skin wash solutions and gauze as well as the total nonabsorbed amount (*n* = 20).

Sample	m (µg)	Percentage
skin wash solutions	12.1 ± 5.0	13 ± 5%
gauze	3.22 ± 1.70	3.3 ± 1.8%
total nonabsorbed	15.4 ± 6.4	16 ± 7%

**Table 2 pharmaceutics-14-02379-t002:** STL amount and percentage of the applied dosage (961 µg (PK1) or 1048 µg (PK2)) in skin wash solutions and skin extracts.

Sample	PK1 (*n* = 8)		PK2 (*n* = 5)	
	m (µg)	Percentage	m (µg)	Percentage
skin wash solutions	74.4 ± 29.6	7.7 ± 3.1%	43.7 ± 10.9	4.2 ± 1.0%
skin	0.054 ± 0.019	0.006 ± 0.002%	0.392 ± 0.053	0.037 ± 0.005%

**Table 3 pharmaceutics-14-02379-t003:** Bodyweight development of the Arnica (*n* = 6) and control (*n* = 4) groups.

Bodyweight (g)	TD 0	PTD 8	PTD 14
Arnica group	22.5 ± 3.3	22.4 ± 2.8	23.5 ± 3.1
Control group	21.7 ± 1.9	22.4 ± 1.8	23.0 ± 1.4

**Table 4 pharmaceutics-14-02379-t004:** Bodyweight development in the Arnica group (*n* = 10) and the control group (*n* = 6).

Bodyweight (g)	TD 0	TD 21	PTD 14
Arnica group	19.5 ± 2.3	22.1 ± 2.4	23.2 ± 2.5
Control group	19.7 ± 3.0	21.6 ± 3.0	22.9 ± 3.2

**Table 5 pharmaceutics-14-02379-t005:** Blood parameters in mice exposed to Arnica tincture (*n* = 10) and the control group (*n* = 6). Means ± standard deviations of each parameter determined before treatment (TD 0) and on day 8 after treatment (PTD 8). Reference values from [21].

Parameter	Reference	Arnica Group		Control Group	
		TD 0	TD 8	TD 0	TD 8
ALT (U/L)	26–77	45 ± 7	46 ± 5	39 ± 8	35 ± 5
AP (U/L)	45–222	102 ± 9	112 ± 13	105 ± 9	110 ± 5
creatinine (mg/dL)	0.20–1.0	0.30 ± 0.00	0.31 ± 0.07	0.31 ± 0.01	0.32 ± 0.06
BUN (mg/dL)	35–60	51 ± 3	51 ± 7	45 ± 6	46 ± 5
Hb (g/dL)	12–17	15 ± 2	16 ± 1	16 ± 0	17 ± 1
Hto (%)	35–53	52 ± 2	53 ± 2	52 ± 2	56 ± 1
RBC (10^9^/mL)	7.8–11	8.5 ± 1.7	8.5 ± 0.8	8.0 ± 0.3	9.0 ± 0.9
WBC (10^6^/mL)	5.0–12	7.0 ± 0.7	7.0 ± 0.8	8.5 ± 1.8	8.4 ± 0.4

## Data Availability

The data reported in this study are contained within the article. The underlying raw data are available on request from the corresponding author.

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
