# Peer review of "Evaluation of Pharmacokinetic and Toxicological Parameters of Arnica Tincture after Dermal Application In Vivo"

_pharmaceutics, 2022, doi:10.3390/pharmaceutics14112379_

Round 1
Reviewer 1 Report
This study highlights the possibility of using Arnica tincture for treatment of Cutaneous leishmaniasis (CL). The authors studies pharmacokinetic properties of the bioactive, antileishmanial compounds and therefore they perfomed in vivo absorption studies of the bioactive compounds after dermal application of Arnica tincture with rats. The authors also looked at the bicompatability of Arnica tincture by performing tests for corrosive or irritant activity as well as acute and repeated toxicity. The results indicated that the compound was biocompatible. The low STL concentration observed in plasma, urine, and feces also show low toxicity using Arnica tincture. Therefore, this compound is a viable candidate for further exploration in future clinical trials for treatment of CL.
Author Response
Reviewer 1
This study highlights the possibility of using Arnica tincture for treatment of Cutaneous leishmaniasis (CL). The authors studies pharmacokinetic properties of the bioactive, antileishmanial compounds and therefore they perfomed in vivo absorption studies of the bioactive compounds after dermal application of Arnica tincture with rats. The authors also looked at the bicompatability of Arnica tincture by performing tests for corrosive or irritant activity as well as acute and repeated toxicity. The results indicated that the compound was biocompatible. The low STL concentration observed in plasma, urine, and feces also show low toxicity using Arnica tincture. Therefore, this compound is a viable candidate for further exploration in future clinical trials for treatment of CL.
We thank the reviewer for the thorough analysis of our work!
Reviewer 2 Report
The authors evaluated the dermal disposition and safety of active ingredients in Arnica tincture, a well-known pharmaceutical product. Their study used both in vivo models (rats and mice) and an artificial skin model (RhE). The manuscript is exceptionally well-written and clearly structured, and the data presentation is praise-worthy. All the experiments appear to be well-planned, but there are some issues that the authors have to address:
- What was the justification for such extensive use of the animal model for evaluating the safety and tolerability of a tincture that has been available and used for decades? In the eyes of the reviewer, such evaluation is excessive and unnecessary, even though the authors took great care to follow OECD guidelines strictly.
- Although the safety study was a success and (expectedly) proved overall good safety and tolerability profile of Arnica tincture, the PK study failed. The authors could not detect meaningful concentrations of the active ingredients in any studied matrices. After carefully reading their previous report (10.3390/pharmaceutics14040742), the reviewer suspects that the method may be the culprit. To the reviewer's knowledge, UHPLC methods, especially with MS detection, require very low injection volumes. Such volumes were used by the authors (2 ul and 20 ul), but larger volumes were also applied (up to 200 ul). 20 uL is quite substantial for a UHPLC system, but 200 uL may lead to worse peak shapes and worse detection. The other aspect is the volume in which the samples were reconstituted. According to the authors, the process demanded 1 mL of a solvent or a mobile phase to reconstitute the extracts. However, there are some possibilities to use smaller volumes efficiently. For example, conical-shaped vial inserts. With such inserts, the reconstituted volume could be reduced to 100 - 150 uL, leading to an almost 10-fold increase in the STL concentration in the samples. The reviewer thinks there was a space for improvement, and it is a shame that the analysis cannot be repeated.
- According to the authors' previous report, STL is highly bound to skin structures, and up to 7% of STL were found in human and porcine skin extracts. In this study, however, only 0.006% of the administered dose was found after 96h in the skin extracts. Was the extraction procedure efficient enough? Why did the authors choose a methanol wash and sonication and not a harsher method, e.g., homogenization? Could there be a method to "unbind" STL from skin samples?
- A direct comparison of PK of two different compounds (substances present in Arnica tincture and radiolabeled analog of helenalin) should be made with caution. Even deuterated molecules can follow different metabolic and elimination patterns than their "native" cousins.
- Please consider changing the title, as the study failed to evaluate any PK parameters for Arnica tincture applied topically.
- If Arnica tincture active ingredients are quickly metabolized and highly bound by skin structures, the remaining unbound amount might be too small to exert any meaningful pharmacological effect. It would undermine the authors' main notion that the Arnica extract might be useful in treating leishmaniasis.
Author Response
Reviewer 2
The authors evaluated the dermal disposition and safety of active ingredients in Arnica tincture, a well-known pharmaceutical product. Their study used both in vivo models (rats and mice) and an artificial skin model (RhE). The manuscript is exceptionally well-written and clearly structured, and the data presentation is praise-worthy. All the experiments appear to be well-planned, but there are some issues that the authors have to address:
- What was the justification for such extensive use of the animal model for evaluating the safety and tolerability of a tincture that has been available and used for decades? In the eyes of the reviewer, such evaluation is excessive and unnecessary, even though the authors took great care to follow OECD guidelines strictly.
The objective to use Arnica tincture in a new field of indication (treatment of cutaneous Leishmaniasis), outside the established traditional use, renders it necessary to carefully reassess these characteristics. In the first clinical study which is shortly to begin, the tincture will be applied in undiluted form to the patients’ skin lesions, which is different from the recommended use in traditional indications, where the tincture is to be diluted 3-10 fold. We think that this is justification enough for the present study.
- Although the safety study was a success and (expectedly) proved overall good safety and tolerability profile of Arnica tincture, the PK study failed. The authors could not detect meaningful concentrations of the active ingredients in any studied matrices. After carefully reading their previous report (10.3390/pharmaceutics14040742), the reviewer suspects that the method may be the culprit. To the reviewer's knowledge, UHPLC methods, especially with MS detection, require very low injection volumes. Such volumes were used by the authors (2 ul and 20 ul), but larger volumes were also applied (up to 200 ul). 20 uL is quite substantial for a UHPLC system, but 200 uL may lead to worse peak shapes and worse detection. The other aspect is the volume in which the samples were reconstituted. According to the authors, the process demanded 1 mL of a solvent or a mobile phase to reconstitute the extracts. However, there are some possibilities to use smaller volumes efficiently. For example, conical-shaped vial inserts. With such inserts, the reconstituted volume could be reduced to 100 - 150 uL, leading to an almost 10-fold increase in the STL concentration in the samples. The reviewer thinks there was a space for improvement, and it is a shame that the analysis cannot be repeated.
The analytical method used is rather sensitive considering the low limits of detection for the STLs. However, we do not think that an improvement in the sample workup could have increased the sensitivity of the overall method very much. We agree with the reviewer that large volume injections can lead to bad peak shapes if the sample is dissolved in a solvent mixture with high organic content like 100% methanol. Therefore, we used water/acetonitrile (95/5, v/v) and carefully checked the peak shapes, the detection limits and linearity by adding defined amounts of the analytes to blank samples. After sample preparation and reconstitution in water/acetonitrile (95/5, v/v) peak shapes, detection limits and linearity were fine. For this reason, we think that dissolving the sample in 1000 µl and injecting 200 µl is equivalent to dissolving the sample in 100 µl and injecting 20 µl in this specific case. We decided to use 1.0 mL as final sample volume to have the possibility to repeat the analysis in case of problems in the run.
We certainly agree that the method could be further improved but this would require the use of a more sensitive mass spectrometer such as triple quad MS which was not available to us. We have added a short statement on this issue to the conclusions.
- According to the authors' previous report, STL is highly bound to skin structures, and up to 7% of STL were found in human and porcine skin extracts. In this study, however, only 0.006% of the administered dose was found after 96h in the skin extracts. Was the extraction procedure efficient enough? Why did the authors choose a methanol wash and sonication and not a harsher method, e.g., homogenization? Could there be a method to "unbind" STL from skin samples?
In our previous report, 7% of STL were found in the skin extracts after 48 h of application. In this experiment we could show the continuous resorption of the analytes through the skin into the receptor fluid which mimics the blood. In the current experiment 0.006% of the administered dose was found after 96 h of application. This amount seems plausible if we take into account that the absorption continued for another 48 h, so that another significant portion could be absorbed during this additional time. The extraction procedure in the two compared experiments was identical.
The skin samples (approx. 20 cm²) were cut in small pieces of 1.0-1.5 cm² before extraction with methanol. We chose this extraction procedure to compare the results of the ex vivo absorption study with the in vivo absorption study. For ex vivo absorption studies this is a common method that was recommended to us by a specialized skin laboratory carrying out very many such analyses (see Acknowledgement to our previous paper).
To date, unfortunately, we don’t have an option to “unbind” STL from skin samples. It is likely that this would have to be a method breaking a covalent (thioether) bond as we expect the STLs to be bound covalently after Michael addition to Cystein residues of skin proteins (as mentioned in our previous communication on ex vivo dermal absorption).
- A direct comparison of PK of two different compounds (substances present in Arnica tincture and radiolabeled analog of helenalin) should be made with caution. Even deuterated molecules can follow different metabolic and elimination patterns than their "native" cousins.
We added a short statement mentioning this (see Conclusions) and we thank the reviewer for this remark.
- Please consider changing the title, as the study failed to evaluate any PK parameters for Arnica tincture applied topically.
It is true that we were not able to measure explicit values for the compounds’ concentrations in plasma, urine and feces but their absorption into the skin could be quantified and maximal systemic concentration values could be deduced from the negative results and the detection/quantification limits. We do consider these results as pharmacokinetic parameters and hence do not agree that it may be necessary to change the title. We have therefore left the title unchanged, but if the editor finds this a mandatory change, we will do so. In that case, please let us know.
- If Arnica tincture active ingredients are quickly metabolized and highly bound by skin structures, the remaining unbound amount might be too small to exert any meaningful pharmacological effect. It would undermine the authors' main notion that the Arnica extract might be useful in treating leishmaniasis.
It is not only a notion that the tincture is useful in treating cutaneous Leishmaniasis, but the efficacy has been proven experimentally, please see our publications on the therapeutic effects (10.3390/ph15070776 and 10.3390/molecules23010150). The reviewer’s argument hence contains an error of logic. The interpretation must rather be that the small amount of unbound STLs probably suffices to lead to the very promising efficacy.
We thank this reviewer for the detailed assessment, thoughtful remarks and constructive criticism!
Reviewer 3 Report
The manuscript submitted for consideration is a logical continuation of the work carried out by the authors for several years. It is dedicated to the in vivo study of Pharmacokinetic and Toxicological Parameters of Arnica Tincture after Dermal Application.
The subject of the study is certainly interesting to readers of the journal Pharmaceutics
The study was carried out at a high experimental level, the data obtained are statistically significant.
Before accepting the article for publication, I would like to clarify the following points:
1) In experiments with gauze in section 3.1. I would like to clarify whether a preliminary experiment "introduced-found" were conducted. Does the gauze always absorb the same amount of the drug?
2) I recommend the authors to describe in more detail the method of sample preparation before chromatographic analysis
Author Response
Reviewer 3
The manuscript submitted for consideration is a logical continuation of the work carried out by the authors for several years. It is dedicated to the in vivo study of Pharmacokinetic and Toxicological Parameters of Arnica Tincture after Dermal Application.
The subject of the study is certainly interesting to readers of the journal Pharmaceutics
The study was carried out at a high experimental level, the data obtained are statistically significant.
Before accepting the article for publication, I would like to clarify the following points:
1) In experiments with gauze in section 3.1. I would like to clarify whether a preliminary experiment "introduced-found" were conducted. Does the gauze always absorb the same amount of the drug?
Such experiments were performed by adding a defined volume of Arnica tincture to gauze pieces with a defined area. Subsequently, the gauze pieces were extracted and analyzed with the same method that was applied in the in vivo dermal absorption study described in section 3.1. In this preliminary experiment the deviation of absorbed and extracted active ingredients was only 4%. Nevertheless, the gauzes of each animal were carefully analyzed to determine the exact absorbed content of the active ingredients in the in vivo dermal absorption study.
2) I recommend the authors to describe in more detail the method of sample preparation before chromatographic analysis
Further details on sample preparation prior to chromatographic analysis have been added in section 2.5.
We thank this reviewer for the constructive criticism!
Reviewer 4 Report
The present work is an expected continuation of two previous works (“in vitro” and “ex in vivo”, respectively) of the authors regarding the evaluation of arnica tincture as a new treatment for cutaneous leishmaniasis.
It is very well organized manuscript, very easy to follow and read to the end. The results are well structured, a large number of samples were used for analyzes that led to obtaining conclusive results. The results are also supported by the photos presented in the Supplementary file.
I recommend publishing the paper in its current form
Author Response
Reviewer 4
The present work is an expected continuation of two previous works (“in vitro” and “ex in vivo”, respectively) of the authors regarding the evaluation of arnica tincture as a new treatment for cutaneous leishmaniasis.
It is very well organized manuscript, very easy to follow and read to the end. The results are well structured, a large number of samples were used for analyzes that led to obtaining conclusive results. The results are also supported by the photos presented in the Supplementary file.
I recommend publishing the paper in its current form
We thank this reviewer for the very positive assessment!
Round 2
Reviewer 2 Report
The authors provided a sufficient explanation for all of the Reviewer's remarks. However, the Reviewer still suggests revisiting the UHPLC method and investigating the injection volume and concentration effect. Indeed, the amount of the substance injected into the system would be the same in the example shown by the authors (dissolving the sample in 1000 µl and injecting 200 µl is equivalent to dissolving the sample in 100 µl and injecting 20 µl). But, with this flow rate and small column volume, 200 uL sounds like an overload. Suppose the authors directly compared injecting a small (2 - 5 ul) volume of a concentrated sample with 200 uL of a diluted sample and saw no differences in the performance (peak shape, height). In that case, such conditions should be acceptable. Also, how the authors approached this matter resembles precolumn dilution with large volume injection. This technique should work well for low and medium-polarity compounds. Of note, the desired "column stacking" is obtained when the solvent ratio in the starting gradient is the same as in the sample. In the present study, the starting gradient was 18% ACN, and the sample was reconstituted in 5% ACN.
Author Response
The authors provided a sufficient explanation for all of the Reviewer's remarks. However, the Reviewer still suggests revisiting the UHPLC method and investigating the injection volume and concentration effect. Indeed, the amount of the substance injected into the system would be the same in the example shown by the authors (dissolving the sample in 1000 µl and injecting 200 µl is equivalent to dissolving the sample in 100 µl and injecting 20 µl). But, with this flow rate and small column volume, 200 uL sounds like an overload. Suppose the authors directly compared injecting a small (2 - 5 ul) volume of a concentrated sample with 200 uL of a diluted sample and saw no differences in the performance (peak shape, height). In that case, such conditions should be acceptable. Also, how the authors approached this matter resembles precolumn dilution with large volume injection. This technique should work well for low and medium-polarity compounds. Of note, the desired "column stacking" is obtained when the solvent ratio in the starting gradient is the same as in the sample. In the present study, the starting gradient was 18% ACN, and the sample was reconstituted in 5% ACN.
The UHPLC method was validated according to ICH guideline M10 as reported in our previous work [10.3390/pharmaceutics14040742]. This also included variation of injection volumes, i.e. quality control samples were analyzed by injection of 2 µL and, after 1:100 dilution, of 200 µL. The validation criteria were fulfilled in all cases, i.e. peaks with S/N >3 and S/N >10 were compatible for both injection methods.
The samples were reconstituted in 5% ACN because we also analyzed those of plasma, urine and feces in a method optimized for the analysis of polar metabolites (see our work on this topic in “Metabolites”: 10.3390/metabo12010088). Since neither the native STLs nor any of their metabolites were found, we have not mentioned this other method in the present manuscript to avoid making it more complicated than necessary and we do not think that it would be useful in the absence of other results.
We hope that the reviewer can accept this as an indication that the method was appropriate.
We thank the reviewer again for the thorough work.
